# Comment on Cook, B.; Mulgan, N. Targeted Mop up and Robust Response Tools Can Achieve and Maintain Possum Freedom on the Mainland. *Animals* 2022, *12*, 921

**DOI:** 10.3390/ani13111840

**Published:** 2023-06-01

**Authors:** Mandy Barron, Dean P. Anderson, Grant Norbury, Bruce Warburton

**Affiliations:** Manaaki Whenua Landcare Research, P.O. Box 69040, Lincoln 7640, New Zealand

The recent paper published by Cook and Mulgan [1] describes the effort to detect and eliminate the remaining brushtail possums and the subsequent invading possums in the Perth River Valley, New Zealand, following a large-scale aerial poisoning operation. The authors claim to have achieved the first successful possum elimination over more than 11,000 ha on mainland New Zealand at an unfenced site. Science- and evidence-based management advances via a critical, constructive and robust peer-review process. The fundamental requirement of this process is that study conclusions must be supported with data. We argue that Cook and Mulgan [1] did not provide quantitative evidence to support their claim of possum elimination. This issue risks misleading conservation scientists and managers. We dispute the declaration of successful elimination and the spatial extent of their inference (>11,000 ha).

The authors provide some description of quantitative methods for declaring eradication in Sections 2.4 and 2.5; however, the methods are flawed and were not used to make inferences and support their conclusions. There was no quantitative assessment of possum elimination using the camera network’s surveillance sensitivity [2,3]. In contrast, they provide subjective statements based on belief, such as “… led us to believe there were exactly two animals present” or “Hence, all three possums caught in the Barlow headwaters in 2020 are believed to have been invaders …”.

In fairness, they estimate a “sample detection probability” (*p*; Equation (1)); however, this method is flawed because the denominator is a function of the number of possums they believe are in the area (“exactly two”). Furthermore, this detection probability is not used to make any inference on the probability of elimination, instead being compared to a completely different published metric, i.e., the “single-detector effective sampling area, *a*_0_” [4], which is not a comparable probability of detection. Finally, the estimate to which they refer is sourced from a different detection device (leg-hold traps) to that used in their study (camera traps). There is no quantitative support for the claim of successful elimination.

The authors’ claims on the total area over which elimination was declared and the per hectare costs are misleading. Detection devices were located in less than half of the study area (1 camera per 35 ha × 142 cameras ≈ 4970 ha), which makes it difficult to make inference on the non-surveyed area. Moreover, many cameras appear to be in close proximity to each other and have over-lapping survey areas, which would further reduce the total area surveyed. There is no effort to incorporate the proportion of the area surveyed in the analysis; thus, it is unclear how the authors can claim that possums were eliminated from 11,642 ha when more than half of that area was not surveyed. The failure to take the spatial extent of surveillance into account in a quantitative manner also means that the per hectare costs of elimination and ongoing control are unfairly devalued as they are based on the area initially treated (8659 ha; Table 2), which is approximately double the area under surveillance.

Invasive animal control and elimination is a costly and complicated process and should be guided by robust science. This creates consistent expectations for project costs and outcomes among funders, policy makers, managers, scientists and the public. Publishing elevated claims without empirical support risks misleading interested parties, resulting in wasteful and ineffective invasive animal management programmes. We encourage the authors and the publishing journal to limit conclusions to those that are firmly supported by the data.

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
