# Peer review of "Comment on Cook, B.; Mulgan, N. Targeted Mop up and Robust Response Tools Can Achieve and Maintain Possum Freedom on the Mainland. Animals 2022, 12, 921"

_animals, 2023, doi:10.3390/ani13111840_

Round 1
Reviewer 1 Report
The authors describe three main flaws about Cook and Mulgan (2022): the lacking quantitative analysis, the exaggerated claim of elimination confirmation over a much larger area than the area surveyed, and the underestimation of management costs. I agree with the arguments presented. I also agree with the importance of challenging conclusions in a constructive manner. In this case, it is not only about advancing science but also providing sound recommendations for conservation actions.
Author Response
Thank you for your review. The need to provide “sound recommendations for conservation actions” is what motivated us to write this comment.
Reviewer 2 Report
I read your comment on the document "Targeted Mop up and Robust Response Tools Can Achieve and Maintain Possum Freedom on the Mainland". You wrote "Science and evidence-based management advances via a critical, constructive and robust peer-review process. The fundamental requirement of this process is that study conclusions must be supported by data."
The paper you mention and commented is already been published. It was reviewed before publication. therefore, when you say, that "We dispute the declaration of successful elimination and the spatial extent of their inference", you must provide evidence that the results were different.
In this case, this would mean that you have to provide evidence that opossums were found in the area and at the right time, but that the authors of the article failed to do so or ignored it. Have you provided such evidence? I have not seen it.
Lines 19-24: yes, authors "believe" about the number of animals they registered, and the criticized paper provides reasoning, to why they believe so. Authors used two methods, and to my understanding, usage of the leghold traps presents hard evidence on the possum's presence.
Finalising my review: your comment could be published nearly "as is", if you really have hard evidence possums were present in the area after eradication measures were taken. However, if you can't provide evidence of the presence of possums in the area, I suggest that you either withdraw this publication or change its tone.
Author Response
Thank you for your review but we believe you have missed the point of our comment. The authors of the paper made the claim that they had eliminated possums over more than 11,000 ha, therefore the onus is on them to provide the evidence to support their claims, not us. Our comment details how this published paper has failed to provide adequate evidence to support their claim. Regarding our criticism about belief statements e.g., “…led us to believe there were exactly two animals present”. We were not disputing that the authors had detected two possums, we were challenging their assertion that there were only two (and not more). “Belief” that there were only two is not evidence. Science is not a belief system.